# Bayesian Calibration for Office-Building Heating and Cooling Energy Prediction Model

**Yu Cui, Zishang Zhu \*, Xudong Zhao \*, Zhaomeng Li and Peng Qin**

Center for Sustainable Energy Technologies, Energy and Environment Institute, University of Hull,
Hull HU6 7RX, UK; y.cui-2018@hull.ac.uk (Y.C.); zhaomeng.li-2020@hull.ac.uk (Z.L.); p.qin-2018@hull.ac.uk (P.Q.)
\* Correspondence: zishang.zhu@hull.ac.uk (Z.Z.); xudong.zhao@hull.ac.uk (X.Z.)

**Abstract:** Conventional building energy models (BEM) for heating and cooling energy-consumption prediction without calibration are not accurate, and the commonly used manual calibration method requires the high expertise of modelers. Bayesian calibration (BC) is a novel method with great potential in BEM, and there are many successful applications for unknown-parameters calibrating and retrofitting analysis. However, there is still a lack of study on prediction model calibration. There are two main challenges in developing a calibrated prediction model: (1) poor generalization ability; (2) lack of data availability. To tackle these challenges and create an energy prediction model for office buildings in Guangdong, China, this paper characterizes and validates the BC method to calibrate a quasi-dynamic BEM with a comprehensive database including geometry information for various office buildings. Then, a case study analyzes the effectiveness and performance of the calibrated prediction model. The results show that BC effectively and accurately calibrates quasi-dynamic BEM for prediction purposes. The calibrated model accuracy (monthly CV(RMSE) of 0.59% and hourly CV(RMSE) of 19.35%) meets the requirement of ASHRAE Guideline 14. With the calibrated prediction model, this paper provides a new way to improve the data quality and integrity of existing building energy databases and will further benefit usability.

**Keywords:** building energy model; Bayesian calibration; sensitive analysis; automatic calibration method





## 1. Introduction

With rapid urbanization, building energy consumption has attracted increasing attention globally. Building energy modeling (BEM) is widely used as the most effective method to assess building energy efficiency. Generally, building energy models lack consideration of the urban context and ae always based on over-simplification and strong assumptions, thus leading to a prediction deviation of up to 90% [1]. To improve the confidence level of models, calibration is an essential step in the modeling process.

Currently, two BEM calibration approaches can be adopted: the trial-and-error approach and Bayesian calibration (BC) [2,3]. The trial-and-error approach is an iterative approach to adjusting unknown input parameters and assumptions [4]. It causes a big challenge for the modeler because the accuracy of the model highly relies on the modeler's experience and knowledge in building and modeling. Due to the same reason, this method requires incredible time and effort from both user and computer to achieve an accurate result [5]. Additionally, it is adjusted to match historical energy data, leading to a weakness in forecasting future energy.

The BC method alleviates the problems presented by the trial-and-error method. Instead of adjusting model parameters and assumptions to match the prediction value with measured data, the BC framework attempts to understand the model uncertainty and retain the consistency between the simulation data and measured data [6]. It is an automated calibration technique based on quantifiable evidence rather than experts' intervention,

which can significantly reduce the workload of the modeler [7]. Quality evidence (usually established databases) can provide informative prior distributions for Bayesian inference and initially determine the form of the result of calibrated parameters [8]. In addition, BC can account for all sources of uncertainty, especially inherent uncertainty from the model itself, which is always reflected by other calibration methods. In addition, BC can determine the best estimation of the parameters, making BC better at predicting future real-world observations [9,10]. Additionally, BC alleviates the over-fitting problem because its objective is maximizing the likelihood function of model output, rather than reducing the deviation between model output and measured values [11]. Compared with other common-use training methods (i.e., the Levenberg–Marquardt method), the BC method can more easily achieve a relatively stable model and reduce the possibility of the local minimum solution [12].

There have been further attempts to apply BC to building energy models. The early research focused on applying BC to the building energy model of existing buildings to improve the model representation of the actual building. Its primary purpose was to provide decision support for energy-saving retrofitting [13]. With the same initial purpose, the method was optimized and promoted to a life-cycle analysis by Yuan et al. [14]. This method is effective for individual buildings, and high-temporal-resolution and high-reading-resolution historical energy datasets can support more accurate model calibration [15]. Recently, the application of BC for building energy modeling has been expanding. It has been applied for different building functions (i.e., residential buildings [16], commercial buildings [17]) and different scales of buildings (individual buildings, building clusters [18] and building stocks [19]). Additional focus was put on the statistic performance of different emulators or surrogates under a BC framework [20]. Lim and Zhai [21] compared five meta-models for BC and indicated that the Gaussian-process (GP) model is relatively accurate. In terms of Markov Chain Monte Carlo (MCMC) algorithms, it was found that the no-U-turn sampler is more effective for the Bayesian calibration of building energy models as compared to random walk Metropolis and Gibbs sampling [22].

However, these studies applied BC to refine the output of a dedicated numerical model without the relevant geometry, so that a calibrated model can only retell the energy profile of a specific building/cluster. In many studies, the model is developed for calibrating unknown parameters [23,24], but there is still a lack of research for the purpose of 'prediction'. In recent years, only Chen et al. [25] developed a BC model based on IES-VE building energy software to predict district heat demand. It mainly focuses on material parameters (U-value for different building components) and lacks consideration of the space layouts of the district. Considering the geometry and material information of buildings, this research developed a BC model with generalization ability to be applied to office buildings with different space layouts and materials.

Additionally, poor building-design parametric sources are a significant challenge for BC applications [8]. With the rapid development of energy-monitoring technology and data-storage technology [26], it is gradually possible to record detailed historical energy-consumption data. Still, there are fewer corresponding records for building structural-design parameters, which leads to an imbalance between building structure and energy in the database [27]. Another situation is that the current dynamic BEM method, e.g., Energy-Plus, is usually 'over-parameterized', which means plenty of inter-reliance parameters are compulsory to the model [28]. Scarce supply in databases and excessive demand in models combined to cause a lack of accessibility to the building design parameters.

This paper aims to develop a dedicated BC model based on quasi-dynamic BEM with the ability to deal with diverse inaccurate geometry parameters and material data. This calibrated model is generalized and can be applied to predict the heating and cooling energy consumption of office buildings. The novelty of this research is mainly expressed in the following two points: (1) To improve the generalization ability of the model, this research used a comprehensive dataset including geometry information and material information of 11 various office buildings, taking as its input those parameters generated

by a Bayesian calibrated model with generalization ability. It can not only retell the energy profile of the original buildings (included in the training set) but also predict energy consumption for other buildings (not included in the training set) with their space layout and geometry information. (2) Instead of dynamic simulation tools, this paper investigates the effectiveness of BC in giving results within the ASHRAE required margin when dealing with models with fewer input parameters from the quasi-dynamic simulation method. With the improvement in generalization ability, this calibration simulation method discovered a new way of solving the data quality and integrity issues of the existing building energy databases, which will greatly improve the usability of the existing big-data resource.

In conclusion, this paper shows that the quasi-dynamic simulation method can achieve a similar accuracy level as the dynamic simulation method after BC. It also effectively improves the generalization ability of the calibrated model by using a comprehensive dataset, including geometry information, so that it can be applied to general office buildings. With this improvement, this paper can also contribute to improving the quality and integrity of the existing building energy database.

## 2. Methodology

### 2.1. Research Framework

This paper aims to develop a dedicated BC model based on quasi-dynamic BEM, which can (1) deal with diverse inaccurate geometry parameters and material data, and (2) predict the heating and cooling energy consumption of office buildings. To achieve the objective, this study collected energy data and building information from 10 office buildings located in Guangdong, China. Another building (with the same data structure) is recorded in a separate dataset to carry out a case study for testing the performance and feasibility of BC prediction model. These buildings were selected according to the building scale, including both open and unit office buildings. A total of 8 key parameters (shown in Table 1) were surveyed to generate the building geometry. Two years of hourly historical energy data and corresponding weather data were gathered to form the energy dataset. With the collected information, this paper calibrated quasi-dynamic BEM ISO 13790 [29] with the BC method [6].

The preparation work has three parts:

1.  A pre-calibration primary model based on ISO 13790 was developed;
2.  A comprehensive energy dataset containing simulation data of primary model and real-world measured data;
3.  A dedicated hypothesis for Bayesian inference was made for the proposed calibration.

The main calibration contains 5 steps:

1.  Use sensitivity analysis(SA) to identify sensitive parameters to construct the calibration parameter $\Theta$;
2.  Assume prior probability density function at weakly informative level;
3.  Combine measured data **z** (collected from office buildings in Guangdong, China) and simulation data **y** (generated by ISO 13790) in a Gaussian process (GP) emulator to generate the prior distribution;
4.  Explore the posterior distributions of the calibration parameter $\Theta$, the location parameter $\beta$ and the hyper-parameter $\phi$ by using the Gibbs sampling approach for the MCMC sampling;
5.  Evaluate the performance of the calibrated model for both convergences for multiple MCMC chains and prediction accuracy.

Finally, a case study was carried out to analyze the calibrated model. The whole methodology framework of applying BC to ISO 13790 is shown in Figure 1.

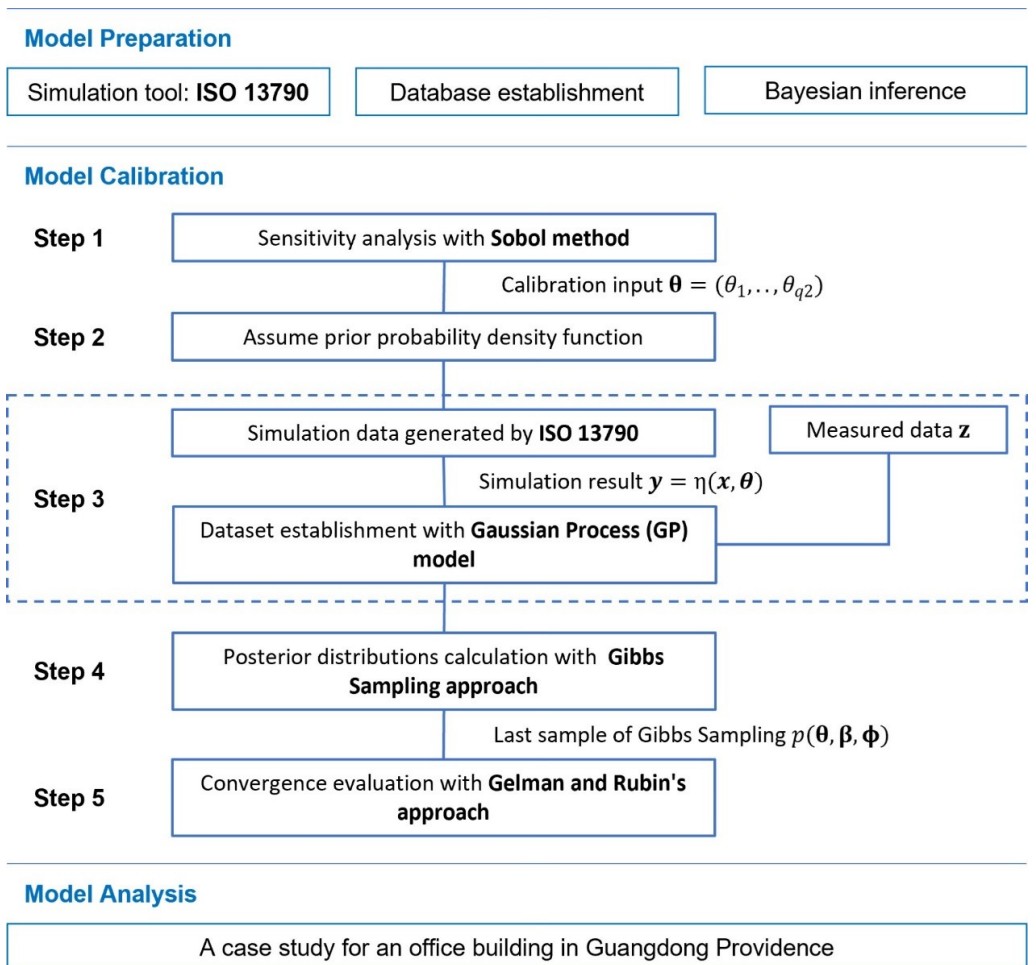

**Figure 1.** BC framework for building energy model.

*2.2. Simulation tool:ISO 13790*

This paper uses ISO 13790 instead of EnergyPlus as a simulation tool to mitigate the problem of over-parameterization and reduce the difficulty of data collection. This step aims to develop a primary energy prediction tool using ISO 13790 and further understand the essential parameters in this model for generating high-quality simulation data. The ISO 13790 calculation method [29] iterates the building thermal mass temperature in specific time steps by modeling the thermal gains and loss balance of the building. It is simulated in an equivalent three-node (surface node, internal air node, and thermal mass node) resistance-capacitance network (5R1C model). The schematic of the 5R1C model is shown in Figure 2.

Based on the schematic of the 5R1C model (Figure 2), the heating/cooling energy loads calculation requires 5 heat transfer rates ($H_{ve}$, $H_{tr,is}$, $H_{tr,w}$, $H_{tr,em}$, and $H_{tr,ms}$), 2 heat gains ($\phi_{sol}$ and $\phi_{int}$) and 4 temperatures ($\theta_{H,set}$, $\theta_{C,set}$, $\theta_{ext}$, and $\theta_{t-1}$). The calculation model for ISO 13790 is shown in Figure 3. In the first part, necessary building properties were used in dataset establishment, and the specific values will be described in Section 2.3, Table 1. In the second part, 19 input variables were used in SA, and relative information will be introduced in Section 2.5, Table 2. In the third part, process variables were involved in the simulation, but these calculation processes can be regarded as black boxes and will not be discussed in the calibration process. The meaning of process variables is shown in Table 3. Finally, the model accessed the simulation result as heating and cooling energy consumption.

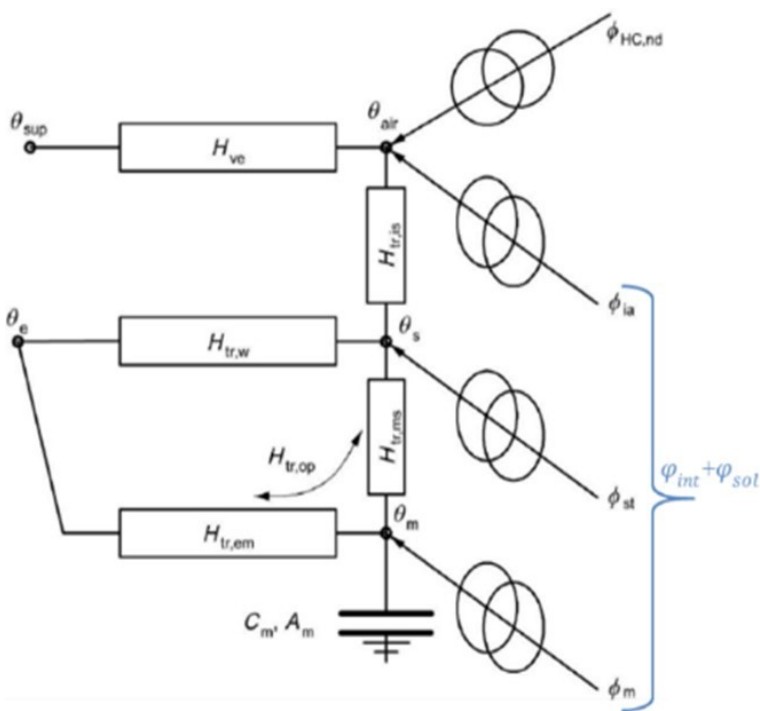

**Figure 2.** The schematic of 5R1C model [29].

### Necessary building properties (Refer to Table 2 for more information and explanation)

| | | | |
|---|---|---|---|
| Gross Floor Area (GFA) | Building Thermal Insulation System | Exterior Wall Material | Heating System |
| Window Framework | Average Window Wall Ratio (WWR) | Glazing Material | Cooling System |

### Input variables (Refer to Table 3 for more information and explanation)

| | | | | | | |
|---|---|---|---|---|---|---|
| **Geometry information** | GFA | L | W | h | WWR | Orie |
| **Material information** | $U_{window}$ | $U_{wall}$ | $U_{roof}$ | | | |
| **Ventilation information** | $Ven_M$ | | | | | |
| **Weather information** | DHI | DNI | $\theta_{ext}$ | $\theta_{t-1}$ | | |
| **Setting point** | $\theta_{C,set}$ | $\theta_{H,set}$ | | | | |
| **Time Serious** | DayNum | HourNum | | | | |
| **Internal heat gain** | $\varphi_{int}$ | | | | | |

### Process variables (Refer to Table 1 for more information and explanation)

| | | | | | |
|---|---|---|---|---|---|
| **Heat transfer** | $H_{ve}$ | $H_{tr,is}$ | $H_{tr,w}$ | $H_{tr,em}$ | $H_{tr,ms}$ |
| **Heat gain** | $\varphi_{int}$ | $\varphi_{sol}$ | | | |
| **Temperature** | $\theta_{C,set}$ | $\theta_{H,set}$ | $\theta_{ext}$ | $\theta_{t-1}$ | |

### Result

## Heating and cooling energy consumption

**Figure 3.** Calculation model for ISO 13790.

**Table 1.** List of necessary building properties for ISO 13790 calculation model. (Necessary building properties in Figure 3).

| Catalogue | Properties |
|---|---|
| GFA | Scale- Range from 5000 to 63,000 m$^2$ |
| Exterior wall material | 'Brick'<br>'Concrete'<br>'Lime–sand brick' |
| Building thermal insulation system | 'Internal insulation'<br>'External insulation' |
| Glazing material | 'Clear'<br>'Coated'<br>'Low-e' |
| Window framework | 'Plain metal window framework'<br>'Heat insulation metal window framework' |
| Window–wall ratio | Scale range from 0.2–0.8 |
| Heating system | 'Radiator'<br>'Air conditioner' |
| Cooling system | 'Centralized air-conditioning system'<br>'Fan coil fresh-air system'<br>'Split air conditioning system' |
| Heating and cooling energy consumption | 2 × 8760 matrix-<br>Two years' hourly energy consumption data |

**Table 2.** List of input variables for ISO 13790 calculation model. (Input variables in Figure 3).

| Num. in 1st SA | Num in 2nd SA | Symbol | Meaning | Unit | Range Max | Range Min |
|---|---|---|---|---|---|---|
| 1 | - | $GFA$ * | Gross floor area | m$^2$ | $1.8 \times 10^5$ | $8 \times 10^3$ |
| 2 | 1 | L | Building length | m | 80 | 15 |
| 3 | 2 | W | Building width | m | 80 | 15 |
| 4 | 3 | h | Story height | m | 3.9 | 3 |
| 5 | 4 | WWR | Average window–wall ratio | - | 0.8 | 0.2 |
| 6 | 5 | Orie | Orientation ¯ | ° | 45 | 0 |
| 7 | 6 | $U_{window}$ | Exterior window U-value | W/m$^2$· K | 6 | 1 |
| 8 | 7 | $U_{wall}$ | Exterior wall U-value | W/m$^2$· K | 1.5 | 0.6 |
| 9 | 8 | $U_{roof}$ | Roof wall U-value | W/m$^2$· K | 1.5 | 0.6 |
| 10 | 9 | $Ven_M$ | Mechanical ventilation | h$^{-1}$ | 6 | 0 |
| 11 | 10 | $DHI$ | Diffuse horizontal irradiance | kW/m$^2$ | 581 | 0 |
| 12 | 11 | $DNI$ | Direct normal irradiance | kW/m$^2$ | 795 | 0 |
| 13 | 12 | $\theta_{ext}$ | External temperature | °C | 37 | 6 |
| 14 | - | $\theta_{t-1}$ * | Last-hour room temperature | °C | 12 | 35 |
| 15 | 13 | $\theta_{H,set}$ | Heating setpoint | °C | 22 | 18 |
| 16 | 14 | $\theta_{C,set}$ | Cooling setpoint | °C | 267 | 22 |
| 17 | 15 | DayNum | Day of year | - | 365 | 1 |
| 18 | 16 | HourNum | Hour of day | - | 23 | 0 |
| 19 | 17 | $\phi_{int}$ | Internal heat gain | W | 20 | 0 |

* is used for the scene classification of SA and is not part of $\mathbf{x}_M$. ¯ The definition of orientation is the angle between normal direction of main façade and the south.

**Table 3.** List of process variables and their meaning for ISO 13790 calculation model. (Process variables in Figure 3).

| Symbol | Meaning |
|---|---|
| $H_{ve}$ | Heat transfer rate by ventilation |
| $H_{tr,is}$ | Heat transfer rate between building surface and internal air |
| $H_{tr,w}$ | Heat transfer rate through the window |
| $H_{tr,em}$ | Heat transfer rate between external air to the building mass |
| $H_{tr,ms}$ | Heat transfer rate between building mass and internal surface |
| $\phi_{sol}$ | Solar heat gain |
| $\phi_{int}$ | Internal heat gain |
| $\theta_{H,set}$ | Heating setpoint |
| $\theta_{C,set}$ | Cooling setpoint |
| $\theta_{ext}$ | External Temperature |
| $\theta_{t-1}$ | Last-hour room temperature |

*2.3. Dataset Establishment*

The entire dataset contains two datasets: measured dataset $\mathbf{D}^m$ and simulation dataset $\mathbf{D}^s$. The measured dataset $\mathbf{D}^m$ is a combination of variable inputs matrix $\mathbf{x}$ and calibration target $\mathbf{z}$, including the building design information and corresponding energy data of 10 office buildings collected from Guangdong, China. The specific index is shown in Table 1.

Variable inputs matrix $\mathbf{x} = (x_1, x_2, \cdots, x_{q_1})$, which contains '$q_1$' non-sensitive parameters, was determined by collecting building design information. The calibration target defined as $\mathbf{z} = (z_1, z_2, \cdots, z_n)^T$ has '$n$' measured energy data and each element in $\mathbf{z}$ is one hour measured heating and cooling energy consumption. According to Table 1, for each building, there is a $2 \times 8760$ matrix for each building energy record. With 10 office buildings integrated composing the $\mathbf{D}^m$, the number of '$n$' in vector $\mathbf{z}$ should be $n = 10 \times 2 \times 8760$. For better understanding, data points in $\mathbf{z}$ are illustrated in Figure 4.

The structure of the measured data set $\mathbf{D}^m$ is:

$$\mathbf{D}^m = [\mathbf{x}, \mathbf{z}] = \begin{bmatrix} x_{1,1} & x_{2,1} & \cdots & x_{q_1,1} & z_1 \\ x_{1,2} & x_{2,2} & \cdots & x_{q_1,2} & z_2 \\ \vdots & \vdots & \ddots & \vdots & \vdots \\ x_{1,n} & x_{2,n} & \cdots & x_{q_1,n} & z_n \end{bmatrix} \tag{1}$$

The simulation dataset $\mathbf{D}^s$ also has three parts: the variable inputs matrix $\mathbf{x}$, calibration input $\Theta$, and corresponding simulation results $\mathbf{y}$. The $\mathbf{x}$ and $\Theta$ come from input variables of the ISO 13790 calculation model, which is divided according to the sensitive analysis result. The variable inputs $\mathbf{x} = (x_1, x_2, \cdots, x_{q_1})$, which include '$q_1$' non-sensitive parameters, will have determined values for each simulation process. The calibration input $\Theta = (\Theta_1, \Theta_2, \cdots, \Theta_{q_2})$, which include '$q_2$' sensitive parameters, will be supposed to take unknown values that need to be calibrated. The simulation result is defined as $\mathbf{y} = (y_1, y_2, \cdots, y_N)$, including '$N$' times simulation runs ($N = 100$ in this study). The mapping relationship between simulation input and result is denoted as $\eta(\mathbf{x}, \Theta)$. Thus, $y_j = \eta(\mathbf{x}_j, \Theta_j)$. The structure of the simulation dataset $\mathbf{D}^s$ is:

$$\mathbf{D}^s = [\mathbf{x}, \boldsymbol{\Theta}, \mathbf{y}] = \begin{bmatrix} x_{1,1} & \cdots & x_{q_1,1} & \Theta_{1,1,1} & \cdots & \Theta_{q_2,1,1} & y_{1,1} \\ x_{1,2} & \cdots & x_{q_1,2} & \Theta_{1,2,1} & \cdots & \Theta_{q_2,2,1} & y_{2,1} \\ \vdots & \ddots & \vdots & \vdots & \ddots & \vdots & \vdots \\ x_{1,n} & \cdots & x_{q_1,n} & \Theta_{1,n,1} & \cdots & \Theta_{q_2,n,1} & y_{n,1} \\ x_{1,1} & \cdots & x_{q_1,1} & \Theta_{1,1,2} & \cdots & \Theta_{q_2,1,2} & y_{1,2} \\ x_{1,2} & \cdots & x_{q_1,2} & \Theta_{1,2,2} & \cdots & \Theta_{q_2,2,2} & y_{2,2} \\ \vdots & \ddots & \vdots & \vdots & \ddots & \vdots & \vdots \\ x_{1,n} & \cdots & x_{q_1,n} & \Theta_{1,n,2} & \cdots & \Theta_{q_2,n,2} & y_{n,2} \\ & & \vdots & & & \vdots & \vdots \\ x_{1,1} & \cdots & x_{q_1,1} & \Theta_{1,1,N} & \cdots & \Theta_{q_2,1,N} & y_{1,N} \\ x_{1,2} & \cdots & x_{q_1,2} & \Theta_{1,2,N} & \cdots & \Theta_{q_2,2,N} & y_{2,N} \\ \vdots & \ddots & \vdots & \vdots & \ddots & \vdots & \vdots \\ x_{1,n} & \cdots & x_{q_1,n} & \Theta_{1,n,N} & \cdots & \Theta_{q_2,n,N} & y_{n,N} \end{bmatrix} \tag{2}$$

Therefore, the dataset to be analyzed is a combination of $\mathbf{y}$ and $\mathbf{z}$: $\mathbf{D}^T = [\mathbf{y}^T, \mathbf{z}^T]$.

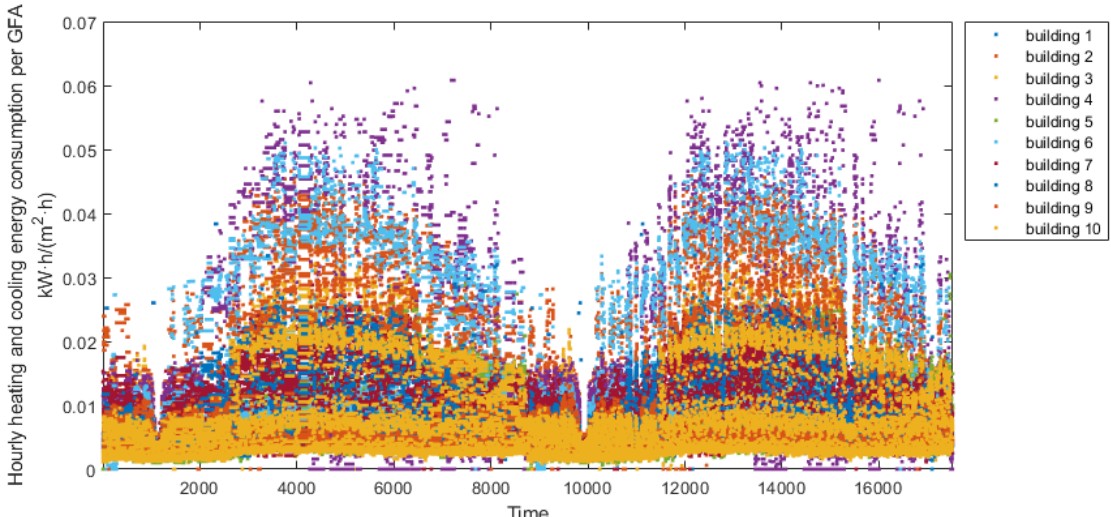

**Figure 4.** Illustration of 2 years' hourly heating and cooling energy consumption per building area for 10 buildings in Guangdong, China.

### 2.4. Bayesian Inference

BC [6] represents all unknown inputs as the parameter $\Theta$. The posterior distribution of $\Theta$ is calculated using the measured data, and the uncertainty of $\Theta$ is quantified. Then, the calibrated model can obtain the prediction distribution based on the posterior distribution of $\Theta$. The preparation of calibration includes: (1) model hypothesis and (2) model analysis.

#### 2.4.1. Model hypothesis

Based on Kennedy and O'Hagen's formulation, the uncertainty of ISO 13790 can come from four aspects: parameter uncertainty, model inadequacy, residual variability, and observation error. The relationship between $y = \eta(\mathbf{x}, \boldsymbol{\Theta})$ and $\mathbf{z}$ can be represented as Equation (3):

$$z_i = \zeta(\mathbf{x}_i) + e_i = \rho\eta(\mathbf{x}_i, \boldsymbol{\Theta}) + \delta(\mathbf{x}_i) + e_i \tag{3}$$

where, $\zeta(\mathbf{x}_i)$ is the true process for $i^{th}$ set of variable input $\mathbf{x}$;
$e_i$ is the observation error for $i^{th}$ sample;
$\rho$ is the regression parameter of simulation model;
$\delta(\mathbf{x}_i)$ is the model inadequacy function, which is independent of the simulation result $\eta(\mathbf{x}_i, \boldsymbol{\Theta})$.

### 2.4.2. Model Analysis

The observation error $e_i$, which considers the residual variability as well, can be assumed as an independent normal distribution $\mathcal{N}[0, \lambda_e]$ ($\lambda_e$ is an unknown variable which is one of the hyperparameters that needs to be calculated in the next step). It is not related to the variable input $\mathbf{x}$, and, further, it is equal for the true process $\zeta(\mathbf{x}_i)$ and simulation process $\eta(\mathbf{x}_i, \Theta)$, so that Equation (3) can be simplified as:

$$\zeta(\mathbf{x}_i) = \rho\eta(\mathbf{x}_i, \Theta) + \delta(\mathbf{x}_i) \tag{4}$$

As in Equation (3), $\eta(\mathbf{x}_i, \Theta)$ and $\delta(\mathbf{x}_i)$ are independent in (4).

### 2.5. Step 1: Sensitive Analysis

Under the ideal situation, all the uncertain parameters in the BEM calculation model should be calibrated [20]. However, due to the data quantity/quality limitation, it is a common practice to identify the uncertain parameters using the SA technique [30]. Tian [31] conducted a comprehensive review of various SA strategies in BEM and stated the input parameters and their range should be defined based on the research purpose. This study chose Sobol method [32] for ISO 13790 to figure out which parameters need to be calibrated. Assuming that all inputs are independently and uniformly distributed within their range, the ranges of input variables are determined according to the measured building properties. The model of ISO 13790 is taken as a black box function with a '$d$' dimensions input vector $\mathbf{X}_M$ and a scaled output $\mathbf{Y}_M$ (the subscript M means model).

A total of 19 input variables for the ISO 13790 calculation model and their range are shown in Table 2. At the first attempt, the SA was conducted with '$d = 19$'. For a more detailed analysis and further clarification of the influence of each variable, the two most influential parameters $GFA$ and $\theta_{t-1}$ were used to divide the situation. The SA was again conducted for different building scales ($GFA$ = 10 k, 50 k, 100 k, 150 k (m$^2$)) and different initial temperatures ($\theta_{t-1}$ = 10, 20, 30 (°C)), respectively. Therefore, the dimensions of input vector $\mathbf{X}_M$ is 17 ($d = 17$) for 2nd SA.

The sensitivity of input parameters was estimated by first-order indices $S_i$ and total-effect index $S_{Ti}$, which is calculated by the following equations:

First-order indices:

$$S_i = \frac{Var_{X_i}(E_{X_{\sim i}}(Y|X_i))}{Var(Y)} \tag{5}$$

Total-effect index:

$$S_{Ti} = \frac{E_{X_{\sim i}}(Var_{x_i}(Y|X_{\sim i}))}{Var(Y)} \tag{6}$$

where,

$X_i$ is $i^{th}$ elements in the vector $X_M$;

$X_{\sim i}$ is all elements in $X_M$ except $X_i$;

$Y$ is the heating and cooling energy consumption calculated by the input vector $X_M$.

With the (quasi) Monte Carlo method, both indices can be estimated with generated sample matrix [33]. Two $N \times d$ sample matrix $X_A$ and $X_B$ were firstly developed with Sobol sequence. Then, '$d$' $N \times d$ sample matrix $X_{ABi}$ (with $i = 1, 2, \cdots, d$) were built with $ith$ column of $X_B$, and $X_A$ without $ith$ column. Correspondingly, $Y_A$, $Y_B$, and $Y_{ABi}$ were calculated by running ISO 13790.

For $S_i$ calculation, $Var_{x_i}(E_{X_{\sim i}}(Y|X_i))$ can be estimated as:

$$Var_{X_i}(E_{X_{\sim i}}(Y|X_i)) \approx \frac{1}{N}\sum_{j=1}^{N} Y_{Bj}(Y_{AB^i j} - Y_{Aj}) \tag{7}$$

For $S_{Ti}$ calculation, $E_{X_{\sim i}}(Var_{X_i}(Y|X_{\sim i}))$ can be estimated as:

$$E_{X_{\sim i}}(Var_{X_i}(Y|X_{\sim i})) \approx \frac{1}{2N}\sum_{j=1}^{N}(Y_{Aj} - Y_{AB^i j})^2 \tag{8}$$

The $S_i$ and $S_{Ti}$ were compared with the threshold, and the parameters exceeding the threshold were defined as sensitive parameters which need to be calibrated.

### 2.6. Step 2: Assume Prior Distributions of Calibrated Parameters

Three different informative levels (non-informative, weakly informative, specific informative) can be used to assume prior distributions. Referring to the available information in the measured dataset, there is no strong evidence supporting that calibrated parameter is close to a specific value. Therefore, weakly informative priors (a normal distribution with a standard deviation of 0.2) are recommended rather than highly constrained specific informative priors [34]. These prior probability distributions were defined according to the normalized range [0, 1] of the calibration parameters $\Theta$. The mean value of the prior distribution is determined by the average value of normalized calibrated variables.

### 2.7. Step 3: Prior Distribution (Gaussian-Process Emulator)

A Gaussian process based on a multivariate normal vector was adopted to denote $\eta(\mathbf{x}_i, \Theta)$ and $\delta(\mathbf{x}_i)$ as two normal distributions. The parameters of the multivariate Gaussian distribution are mean value vector $\mathbf{m}$ and covariance function $\mathbf{c}$.

Adopting the linear model with weak prior distribution, $\mathbf{m}$ can be expressed as $\mathbf{m}_\eta = \mathbf{h}_\eta{}^T\boldsymbol{\beta}_\eta$ and $\mathbf{m}_\delta = \mathbf{h}_\delta^T\boldsymbol{\beta}_\delta$, with $p(\boldsymbol{\beta}_\eta, \boldsymbol{\beta}_\delta) \propto 1$. For brevity, $\boldsymbol{\beta}_\eta$ and $\boldsymbol{\beta}_\delta$ were combined in location parameters $\boldsymbol{\beta} = (\boldsymbol{\beta}_\eta^T, \boldsymbol{\beta}_\delta^T)^T$

In terms of covariance function $\mathbf{c}$, this can be calculated by:

$$\mathbf{c}_{\eta,ij} = \frac{1}{\lambda_\eta}exp\{-\sum_{k=1}^{q_1}\beta_{\eta,k}(x_{ik} - x_{jk})^2 - \sum_{k'=1}^{q_2}\beta_{\eta,q_1+k'}(\Theta_{ik'} - \Theta_{jk'})^2\} \tag{9}$$

and,

$$\mathbf{c}_{\delta,ij} = \frac{1}{\lambda_\delta}exp\{-\sum_{k=1}^{q_1}\beta_{\delta,k}(x_{ik} - x_{jk})^2\} \tag{10}$$

Then, the covariance function of the $\mathbf{z}$ can be specified as follows:

$$\mathbf{c}_z = \mathbf{c}_\eta + \begin{bmatrix} \mathbf{c}_\delta + \mathbf{c}_e & 0 \\ 0 & 0 \end{bmatrix}, \text{ in which } \mathbf{c}_e = I_e/\lambda_e \tag{11}$$

This formulation introduces several unknown hyper-parameters to the calibration and inference process: the precision hyperparameters ($\lambda_\eta$ and $\lambda_\delta$), and two sets of correlation hyperparameters, ($\boldsymbol{\beta}_\eta$ and $\boldsymbol{\beta}_\delta$). Then, parameters related to the covariance functions ($\lambda_\eta$, $\lambda_\delta$, $\boldsymbol{\beta}_\eta$ and $\boldsymbol{\beta}_\delta$) were combined and denoted as hyperparameter $\boldsymbol{\varphi}$.

$$L(\mathbf{d}|\Theta, \boldsymbol{\beta}, \lambda) \propto (\mathbf{c}_z)^{\frac{1}{2}}exp\{-\frac{1}{2}(z - \mu)^T(\mathbf{c}_z)^{-1}(z - \mu)\} \tag{12}$$

All hyperparameters ($\rho$, $\lambda$ and $\boldsymbol{\varphi}$) are combined as hyperparameters $\boldsymbol{\phi}$. Therefore, the complete set of parameters comprises the calibration parameter $\Theta$, the location parameter $\boldsymbol{\beta}$ and the hyperparameter $\boldsymbol{\phi}$. It is a reasonable expectation that the calibration parameter $\Theta$ is independent of others, so the prior distribution is:

$$p(\Theta, \boldsymbol{\beta}, \boldsymbol{\phi}) = L(\mathbf{d}|\Theta, \boldsymbol{\beta}, \boldsymbol{\phi}) \times p(\Theta)p(\boldsymbol{\phi}) \tag{13}$$

### 2.8. Step 4: Posterior Distribution (Gibbs Sampling Approach)

To solve the multivariate distribution, the MCMC algorithm was used to compute the probability density function of the calibration parameters considered. Gibbs sampling approach is one of the MCMC algorithms. To assess the posterior distribution function $p(\mathbf{d}|\Theta, \beta, \phi)$, a high-dimension Gibbs sampling algorithm was applied to obtain the sample sequence, which will form the Markov chain. As Markov Chain can only move along the axes in the sample space during the second step, the final sample $p(\Theta, \beta, \phi)$ is converged to a common stationary distribution.

### 2.9. Step 5: Evaluation and Validation

The performance of the calibrated model needs to be evaluated and validated. The evaluation criterion is the matching degree, i.e., accuracy and convergence, between the model predicted energy consumption and the actual measured energy consumption.

#### 2.9.1. Evaluation of Convergence for Multiple MCMC Chains

Gelman and Rubin's approach [35] was applied to assessing the convergence of multiple MCMC chains. The convergence is indicated by an overestimated scale reduction factor $\hat{R}$. For '$m$' sets of '$n$' simulated observations, the indicator $\hat{R}$ was calculated using between sequence variance $B/n$ and the within sequence variance $W$:

$$\hat{R} = \frac{\hat{V}}{W} = \frac{m+1}{m}\frac{\hat{\sigma}_+^2}{W} - \frac{n-1}{mn} \tag{14}$$

where,

$$\hat{\sigma}_+^2 = \frac{n-1}{n}W + \frac{B}{n} \tag{15}$$

When the $\hat{R}$ is close to 1, the potential for variance $\hat{\sigma}_+^2$ to be further decreased is limited, and each of the '$m$' sets of '$n$' simulated observations are close to the target distribution. For convergence, $\hat{R}$ should be approximately $1 \pm 0.1$.

#### 2.9.2. Validation of Prediction Accuracy

The performance of the model was evaluated by using mean bias error (MBE), coefficient of variation of root mean square error (CV(RMSE)), and mean absolute percentage error (MAPE). A total of 25% of samples in the measured dataset $\mathbf{D}^m$ were originally extracted as the testing dataset and used for validation.

$MBE = \frac{\sum(Y^s - Y^m)}{\sum Y^m}$, is the sum of deviation and reflects the overall over-estimation/under-estimation of the model.

$CV(RMSE) = \frac{\sqrt{\sum(Y^s - Y^m)^2}}{\sum Y^m}$, was determined by the RMSE, which is not affected by the sign of the deviation. It reflects the variability in agreement between the simulated results and measured values.

$MAPE = \frac{1}{n} \times \sum \left|\frac{Y^s - Y^m}{Y^m}\right|$ reflects the percentage of error for each prediction.

Model performance was evaluated on different time scales (hourly, monthly). For every signal time scale, a dedicated statistical threshold was set in line with the ASHRAE guideline [36], details are shown in Table 4.

**Table 4.** Threshold limits of statistical metrics for model evaluation [36].

| Statistical Metrics | Hourly Prediction | Monthly Prediction |
|---|---|---|
| MBE (%) | ±10 | ±5 |
| CV(RMSE) (%) | 30 | 15 |

## 3. Results

### 3.1. Result of Step 1: Input Variable and Calibration Variable

For the first attempt, SA on 19 input variables in the calculation model (the number of the variables was same as the series shown in Table 2) was conducted with the Sobol method, and the results are shown in Figure 5. It can be seen from Figure 5 that the $S_i$ and $S_{Ti}$ of variable 1 (GFA) and variable 14 ($\theta_{t-1}$) are too large, leading to difficulties in analyzing the sensitivity of the remaining 17 variables. Therefore, GFA and $\theta_{t-1}$ were used to make a reasonable division of the situation and conduct a further SA in detail.

For the remaining 17 parameters shown in Table 2, SA was conducted under 12 cases with different building scales (GFA = 10 k, 50 k, 100 k, 150 k (m²)) and different initial temperatures ($\theta_{t-1}$ = 10, 20, 30 (°C)); the results ($S_i$, and $S_{Ti}$) are shown in Figure 6.

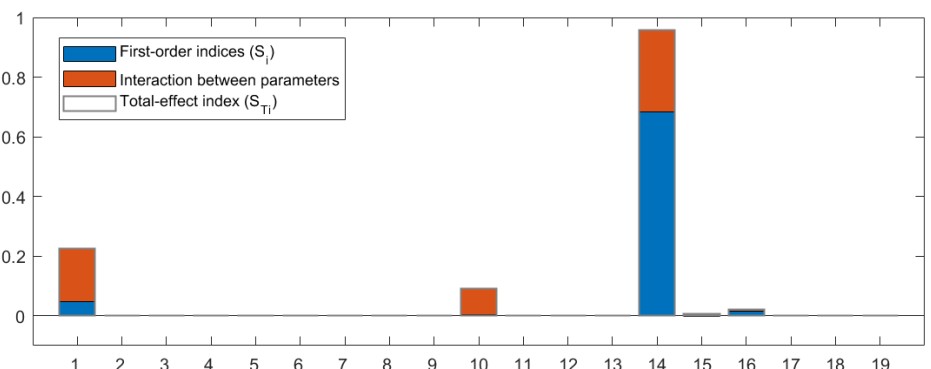

**Figure 5.** Sensitive analysis results for input variables in ISO 13790 calculation model. (Correspondence between numbers and variables can be seen in Table 2—'Num. in 1st SA').

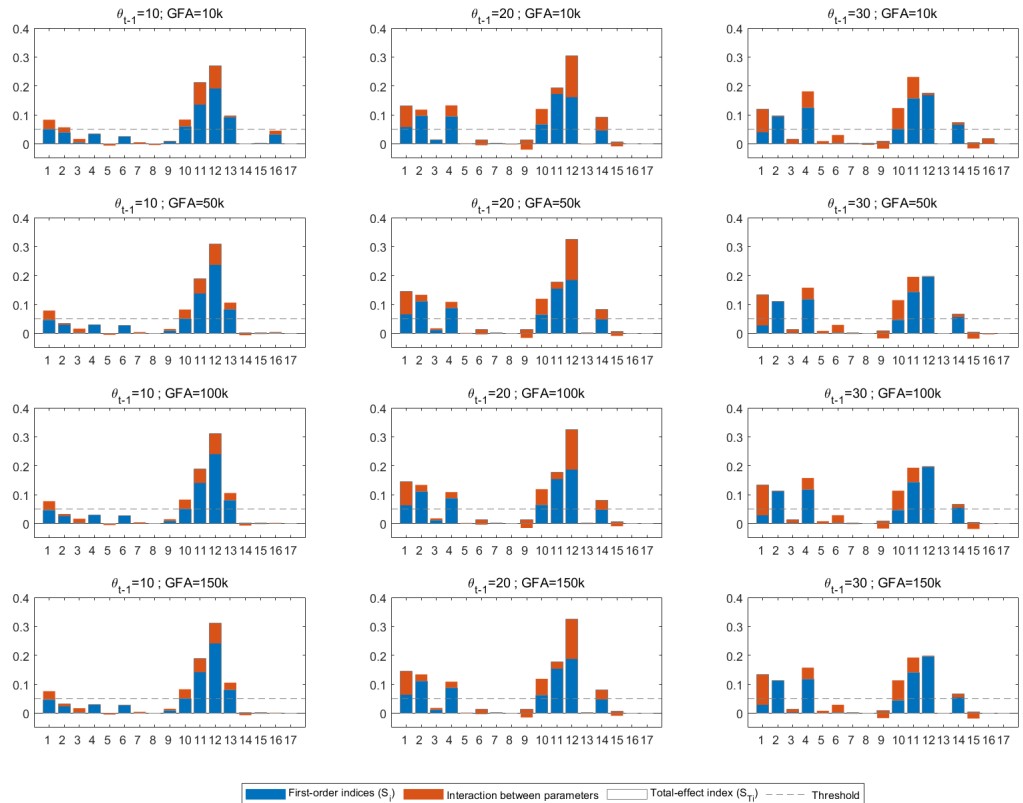

**Figure 6.** Sensitive analysis result for 17 parameters with different GFA and $\theta_{t-1}$. (Correspondence between numbers and variables can be seen in Table 2—'Num. in 2nd SA').

This paper defines the threshold as 0.05 (the dashed line in the figure), and the parameters exceeding the threshold are defined as sensitive parameters. As the GFA of the building can always be achieved as true value, it is not considered as a calibrated variable. Therefore, Last-hour room temperature $\theta_{t-1}$ and parameter 1 (L), 2 (W), 4 (WWR), 10 (DHI), 11 (DNI), 12 ($\theta_{ext}$), 13 ($\theta_{H,set}$), 14 ($\theta_{C,set}$), 16 (HourNum) were, together, defined as $\Theta = (\Theta_1, \Theta_2, \cdots, \Theta_{10})$ for BC. According to the results of SA, 19 input parameters of ISO 13790 were redivided into input variable **x** (with $q_1 = 9$) and calibration variable $\Theta$ (with $q_2 = 10$). Detailed information is displayed in Table 5a and Table 5b, respectively.

**Table 5.** Parameters of input variable **x** and calibration variable $\Theta$.

| (a) Input variable **x** | |
|---|---|
| **Parameter** | **Meaning** |
| $x_1$ | Gross floor area |
| $x_2$ | Day of year |
| $x_3$ | Story height |
| $x_4$ | Exterior window U-value |
| $x_5$ | Exterior wall U-value |
| $x_6$ | Roof wall U-value |
| $x_7$ | Orientation |
| $x_8$ | Mechanical ventilation |
| $x_9$ | Internal heat gain |
| **(b) Calibration variable $\Theta$** | |
| **Parameter** | **Meaning** |
| $\Theta_1$ | Initial temperature |
| $\Theta_2$ | Hour of day |
| $\Theta_3$ | External temperature |
| $\Theta_4$ | Diffuse horizontal irradiance |
| $\Theta_5$ | Direct normal irradiance |
| $\Theta_6$ | Building length |
| $\Theta_7$ | Building width |
| $\Theta_8$ | Average window–wall ratio |
| $\Theta_9$ | Heating setpoint |
| $\Theta_{10}$ | Cooling setpoint |

*3.2. Result of Step 2: Priors Distribution of Model Calibrated Parameters*

The prior distribution functions of calibrated parameters were assumed at the weakly informative level (a normal distribution with a standard deviation of 0.2). These prior probability distributions were defined according to the normalized range [0, 1] of the calibration parameters $\Theta$. The mean value of the prior distribution was determined by the average value of normalized calibrated variables. The detailed information is shown in Table 6.

**Table 6.** Selected prior probability distributions for calibrated parameters.

| Parameter | Meaning | Prior Probability Distribution |
|---|---|---|
| $\Theta_1$ | Initial Temperature | $N(0.412, 0.2)$ |
| $\Theta_2$ | Hour of day | $U(0, 1)$ |
| $\Theta_3$ | External Temperature | $N(0.553, 0.2)$ |
| $\Theta_4$ | Diffuse horizontal irradiance | $N(0.144, 0.2)$ |
| $\Theta_5$ | Direct Normal Irradiance | $N(0.085, 0.2)$ |
| $\Theta_6$ | Length | $N(0.5, 0.2)$ |
| $\Theta_7$ | Width | $N(0.5, 0.2)$ |
| $\Theta_8$ | Average window–wall ratio | $N(0.335, 0.2)$ |
| $\Theta_9$ | Heating setpoint | $N(0.5, 0.2)$ |
| $\Theta_{10}$ | Cooling setpoint | $N(0.5, 0.2)$ |

### 3.3. Result of Step 3: Prior Distributions of Hyperparameters in Gaussian Process Emulator

The hyperparameters of the GP model include: $\beta_{\delta,1}, \beta_{\delta,2}, \cdots, \beta_{\delta,q_1}; \beta_{\eta,1}, \beta_{\eta,2}, \cdots, \beta_{\eta,q_1};$ $\beta_{\eta,q_1+1}, \beta_{\eta,q_1+2}, \cdots, \beta_{\eta,q_1+q_2}; \lambda_\eta; \lambda_\delta; \lambda_e$. The detailed prior distribution assumptions for each hyperparameter are given in Table 7.

**Table 7.** Selected prior probability distributions for hyperparameters of the GP model.

| Parameter | Meaning | Prior Probability Distribution |
|---|---|---|
| $\beta_\delta$ | Correlation strength parameter for model inadequacy | $B(1, 0.4)$ |
| $\beta_\eta$ | Correlation strength parameter for model emulator | $B(1, 0.5)$ |
| $\lambda_\eta$ | Precision parameter for model inadequacy | $\Gamma(10, 0.3)$ |
| $\lambda_\delta$ | Precision parameter for model emulator | $\Gamma(10, 10)$ |
| $\lambda_e$ | Precision parameter for observation error | $\Gamma(10, 0.03)$ |

### 3.4. Result of Step 4: Posterior Distributions of Model Calibrated Parameters

Figure 7 shows the prior and posterior distributions of the 10 calibrate variables. It can be seen that using the weakly informative prior distribution can initially determine the form of the posterior distribution. Meanwhile, the shift in the posterior distribution from the prior distribution indicates that the measured data can affect the posterior distribution. The good combination of information from the prior distribution and measured data enables the prior distribution to exclude unreasonable parameter values, but is not strong enough to exclude meaningful values in the measured dataset.

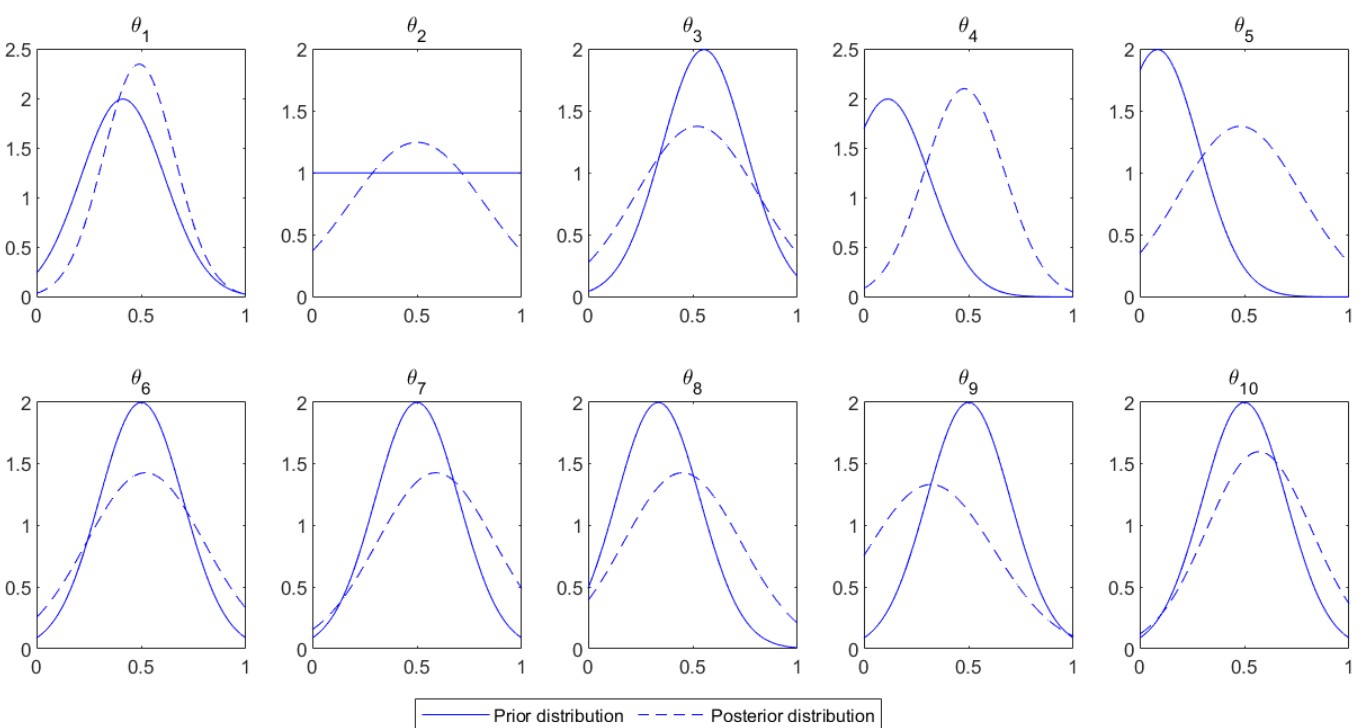

**Figure 7.** The prior and posterior distributions of 10 calibration parameters.

### 3.5. Result of Step 5: Validation of Model Accuracy and Convergence

The Gelman–Rubin statistic $\hat{R}$ was used to check for adequate convergence, and its value should be approximately 1 in this model. $\hat{R}$ values are within $1 \pm 0.1$ for all calibration parameters and hyperparameters of the GP model, which means every parameter in the model is well-converged. Regarding model accuracy validation, the training-set and testing-set results are shown in Table 8. According to the threshold set by ASHRAE Guideline 14, the model can be considered well-calibrated and satisfies the accuracy requirement.

Furthermore, low CV(RMSE) and MBE indicate good consistency between calibration prediction and measurement results.

**Table 8.** Comparison of model performance between before calibration and calibrated model.

| Metrics | Training Dataset | | Testing Dataset | |
|---|---|---|---|---|
| | Monthly | Hourly | Monthly | Hourly |
| MAPE (%) * | 0.26 | - | 0.18 | - |
| MBE (%) | 0.00 | 0.00 | 0.03 | 0.03 |
| CV(RMSE) (%) | 0.59 | 19.35 | 0.79 | 23.50 |

* MAPE metrics cannot be applied for hourly prediction data due to the denominator being 0 for several samples.

## 4. Discussion and Case Study

To demonstrate the performance of the calibrated model, an additional office-building historical energy data was used as a new data set (not included in the previous training set or test set) to carry out a case study. It is an office building with a *GFA* of 63,960 m$^2$. The specific properties of this building are shown in Table 9.

**Table 9.** The properties of the case study office building.

| Catalogue | Properties |
|---|---|
| GFA | 63,960 m$^2$ |
| Exterior wall material | 'Brick' |
| Building thermal insulation system | 'Internal Insulation' |
| Glazing material | 'Clear' |
| Window framework | 'Non-insulated metal' |
| WWR | 0.3 |
| Heating system | 'Radiator' |
| Cooling system | 'Centralized air-conditioning system' |

The overall performance of the calibrated model is shown in Table 10. As can be seen from the table, the monthly CV(RMSE) is 0.73%, and the hourly CV(RMSE) is 20.97%, which is obviously improved from 33.13% and 153.98% in the uncalibrated model. Furthermore, it satisfy the ASHRAE Guideline 14 accuracy requirements.

**Table 10.** Comparison of model performance between before calibration and calibrated model.

| Metrics | Monthly prediction | | | Hourly Prediction | | |
|---|---|---|---|---|---|---|
| | Uncalibrated | Calibrated | Threshold | Uncalibrated | Calibrated | Threshold |
| MAPE (%) * | 30.34 | 0.64 | - | - | - | - |
| MBE (%) | −7.20 | −0.03 | ±5 | −7.20 | −0.03 | ±10 |
| CV(RMSE) (%) | 33.13 | 0.73 | 15 | 153.98 | 20.97 | 30 |

* MAPE metrics cannot be applied for hourly prediction data due to the denominator being 0 for several samples.

Figure 8 compares monthly prediction before and after calibration with the measured data, which can provide a basic understanding of calibration performance. Figure 9 demonstrates the corresponding monthly absolute percentage error (APE) for a more detailed illustration. As shown in Table 10, the MAPE for monthly prediction of the calibrated model is 0.64%, which is calibrated from MAPE 30.34% in the uncalibrated model. In addition, from Figure 9, it can be seen that all APE is within 2.5%, which means the APE performance of the model is stable for every month. It is worth noting that the prediction error for February is slightly higher than that of other months, which can also be seen from Figure 10. It is speculated that the sample number of mixed cooling and heating months is not enough in the sample data, so the calibrated model cannot make a perfectly accurate prediction.

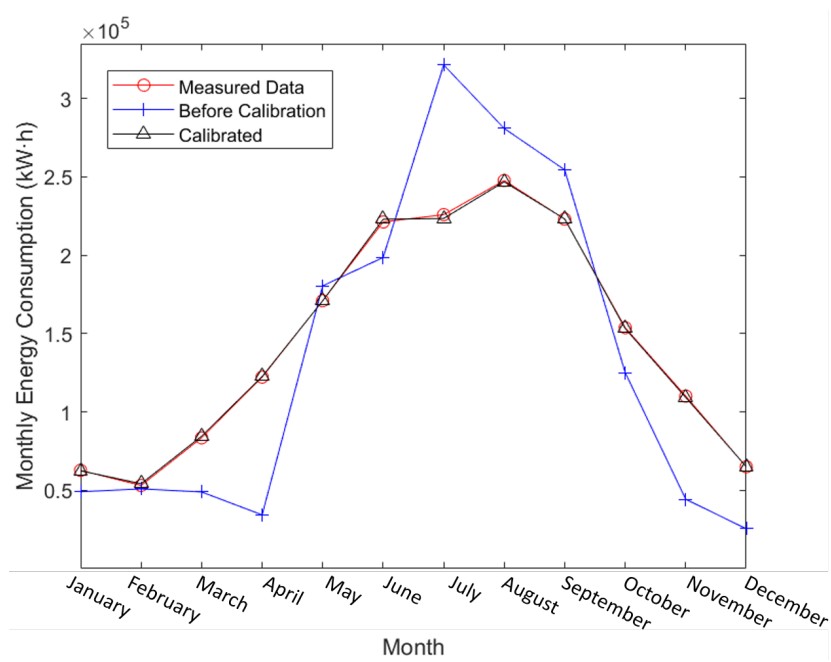

**Figure 8.** Comparison of predicted result and measured data.

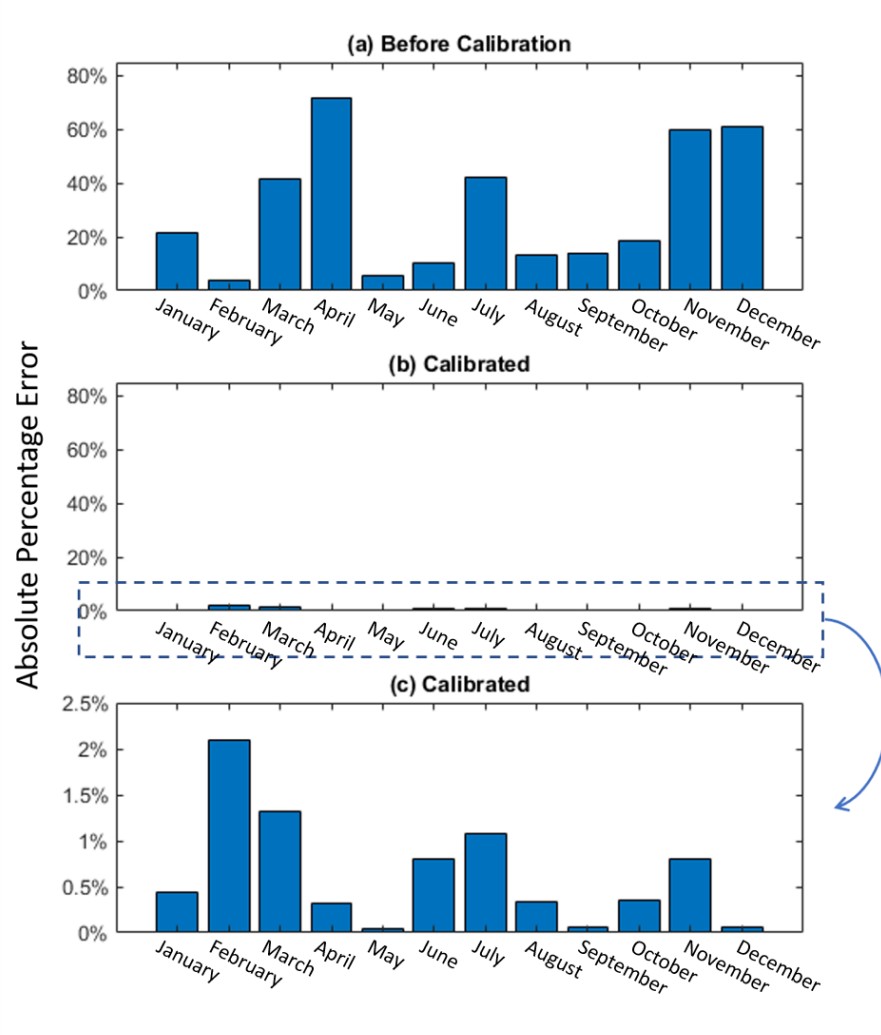

**Figure 9.** Monthly absolute percentage error.

For hourly prediction, the annual average hourly prediction CV(RMSE) is 20.97%, and MBE is −0.03%, which are in line with ASHRAE Guideline 14. The detailed hourly prediction performance in each month is shown in Figure 10.

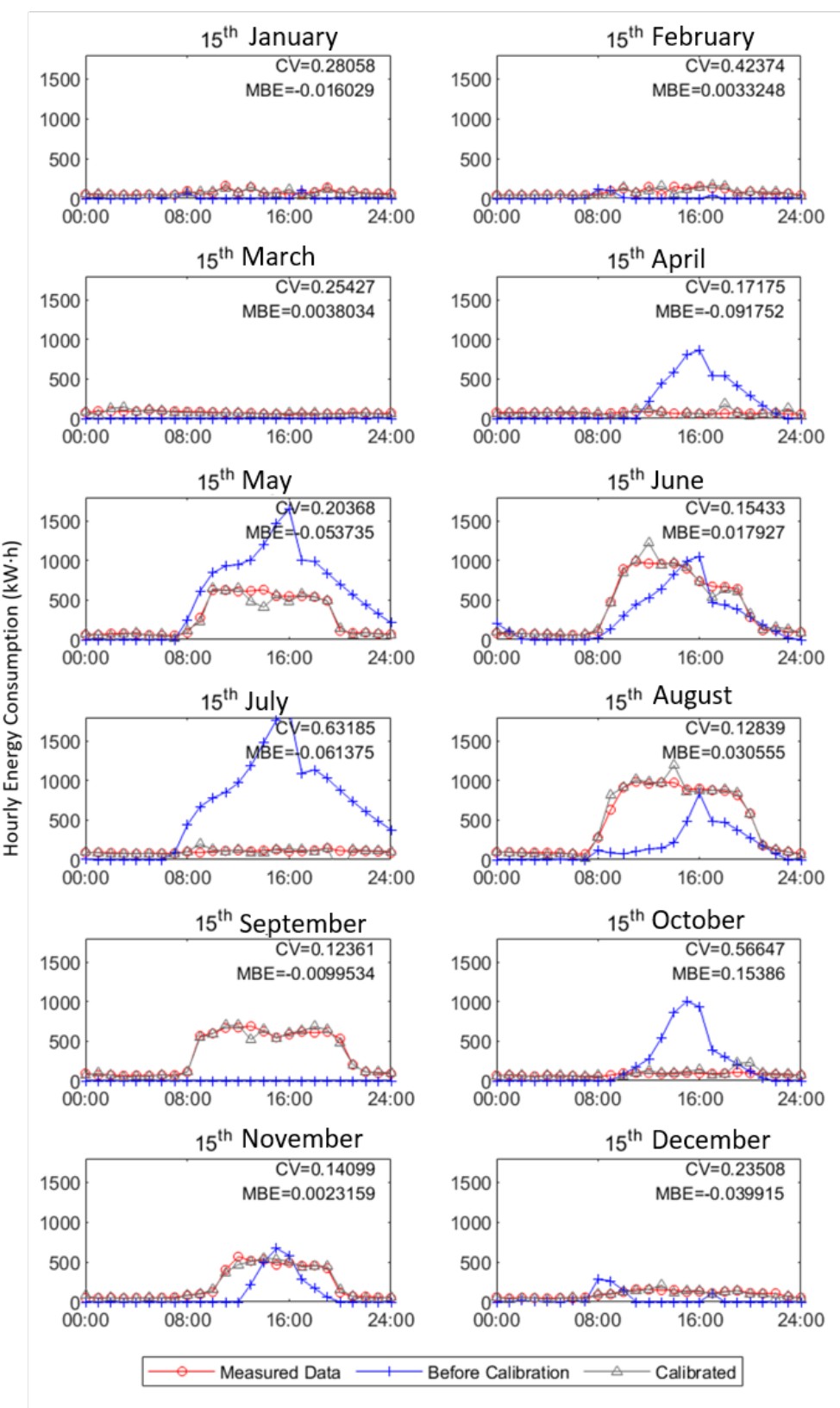

**Figure 10.** Hourly prediction performance over 12 months.

In Figure 10, the hourly prediction CV(RMSE) for all typical days, except 15 February and 15 October, are less than 30%.

The weaker performance in February and October may be attributed to the fast switching of cooling and heating conditions in these two months. The insufficient sample in the dataset leads to the model's simulation accuracy of mixed heating and cooling months being slightly lower than that of purely heating or cooling months.

Figure 11 shows the hourly prediction performance of the calibrated model on design day. In this case study, the winter design day is 21 January, and the summer design day is 21 July. It can be seen from the figure that the calibrated model is very successful in predicting the energy consumption of the summer design day. It successfully corrected the daily maximum and minimum energy consumption and achieved high accuracy with a CV(RMSE) of 5.55%. The prediction of the winter design day also has a large improvement compared with the uncalibrated model, and the overall trend is correct. Although the biggest difference between prediction and measurement is up to 52.39% (at 14:00), the average CV(RMSE) is still within 30% of the day.

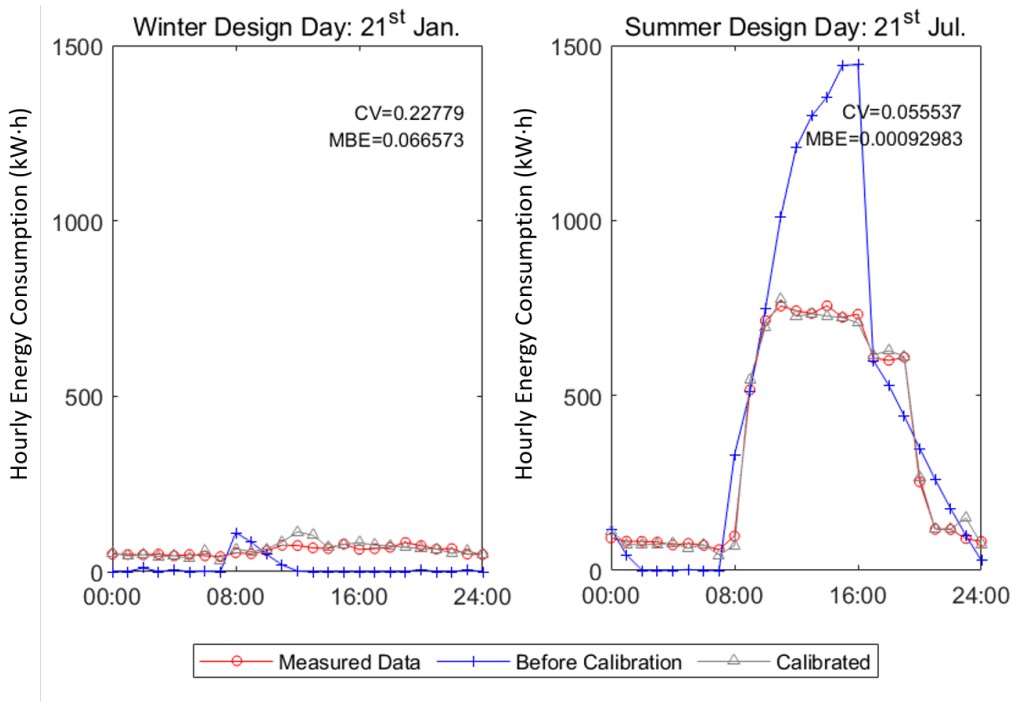

**Figure 11.** Model performance on design day.

Based on the result of the case study, the findings can be summarized as follows: Data set with building energy consumption data combined with building geometry information and material information is useful for BC calibrated prediction model. The calibrated quasi-dynamic simulation method can predict the energy consumption of other buildings not in the training set. The effect of calibration is obvious: it greatly improves the performance of the model for the three metrics (MAPE, MBE, and CV(RMSE)). Meanwhile, both the CV(RMSE) and MBE of the model satisfied the requirement of ASHRAE Guideline 14.

## 5. Conclusions

This paper provides a well-calibrated office-building prediction BEM using the BC method based on the established historical energy dataset, including geometry information, in Guangdong, China. It demonstrates the process of applying the BC to ISO 13790 based on a comprehensive dataset of real energy data from office buildings in Guangdong, China. Based on the result of the case study, this paper also shows the prediction result is in-line with the requirement of the hourly prediction model in ASHRAE Guideline 14.

Step-by-step results and key findings are listed as follows:

1. The Sobol method was used as the SA method. With the threshold of 0.05, 10 of the 19 input values in the ISO 13790 energy calculation method were determined as calibration variables.
2. With less confidence in measured building geometry and material data, the prior probability hypothesis can take a weakly informative level. Taking the weakly informative level prior to distribution can lead to a good combination of information from the prior distribution and measured data, resulting in a proper posterior probability density function with accurate prediction results.
3. A calibrated building energy model was established that meets the accuracy thresholds set by ASHRAE [36], with hourly prediction CV(RMSE) as 20.97% and MBE as $-0.03\%$, which is less than 30% and 10%.
4. The Gelman–Rubin statistic $\hat{R}$ was used to evaluate the model convergence and check the posterior distribution mixing and gathered on a common stationary distribution.

With the previously mentioned findings, the conclusion can be stated. The quasi-dynamic simulation method can achieve a similar accuracy level as the dynamic simulation method after BC. The generalization ability of the calibrated model can be improved with a comprehensive database, including geometry information and material information. The new model can be applied to the office buildings not included in the training set and can satisfy the prediction requirement set by ASHRAE. With this improvement, this calibrated prediction model can also be applied to improve the quality and integrity of the existing building energy database.

**Author Contributions:** Conceptualization, Y.C., Z.Z. and X.Z.; methodology, Y.C., Z.Z. and X.Z.; software, Y.C.; validation, Y.C., Z.L. and P.Q.; formal analysis, Y.C.; investigation, Y.C.; data curation, Y.C. and Z.L.; writing—original draft preparation, Y.C.; writing—review and editing, Y.C., Z.Z., Z.L. and P.Q.; visualization, Y.C. and P.Q.; supervision, Z.Z. and X.Z. All authors have read and agreed to the published version of the manuscript.

**Funding:** This research was funded by the UK Newton Fund and Guangdong Department of Science and Technology OF FUNDER grant number 101005-586174.

**Data Availability Statement:** Not applicable.

**Conflicts of Interest:** The authors declare no conflict of interest.

**Nomenclature**

**Symbols**
| | |
|---|---|
| $\Theta$ | Calibration input matrix |
| $\mathbf{x}$ | Variable inputs matrix |
| $\mathbf{z}$ | Calibration target vector |
| $\phi$ | Heat flow rate/thermal power |
| $\theta$ | Temperature |
| $H$ | Heat transfer coefficients |

**Subscripts**
| | |
|---|---|
| $M$ | Model |
| $C$ | Cooling |
| $H$ | Heating |
| $em$ | Between external air and building mass |
| $ext$ | External |
| $int$ | Internal is between internal air and building surface |
| $ms$ | Between building mass and building surface |
| $set$ | Set point |
| $sol$ | Solar |
| $t-1$ | Last hour |
| $tr$ | Transmission |
| $ve$ | Ventilation |
| $w$ | Through the window |

**Superscripts**

| | |
|---|---|
| $^m$ | Measured |
| $^s$ | Simulation |

**Abbreviations**

| | |
|---|---|
| BC | Bayesian calibration |
| BEM | Building energy model |
| CV(RMSE) | Coefficient of variation of root mean square error |
| GP | Gaussian process |
| MAPE | Mean absolute percentage error |
| MBE | Mean bias error |
| MCMC | Markov chain Monte Carlo |
| SA | Sensitivity analysis |

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
