# Peer review of "Bayesian Calibration for Office-Building Heating and Cooling Energy Prediction Model"

_buildings, doi:10.3390/buildings12071052_

Round 1

Reviewer 1 Report

This article reports the application of the Bayesian method to automate the calibration process of building energy simulation models.

The topic is of great interest to the industry with its potential to save time and resources for calibrated simulations. The logic flow of this article is generally easy to follow.

However, this article requires substantial revision to be considered for publication. Below are my comments:

1.     The authors tend to use strong words such as “prove”, “universal”, but the evidence or argument presented was often not strong enough. I suggest the users soften the language throughout this article.

2.     A common concern for auto-calibration is the fact that the automated process may not reach reliable parameter values in the end. The programme may tend to take unrealistic parameters for the sake of metrics. This is where an expert’s input is still indispensable. This is not addressed by the authors.

3.     Similarly, although the results were validated in terms of statistical metrics, the authors did not present post-calibration parameters of the building.

4.     The authors claim universal generalisability of the method, but I find this unjustified with a training set of eleven buildings and one more test building.

5.     Literature review:

a.     Only 3 out of 23 cited articles are within past 5 years. If after careful search the authors found that there is no recent publication relevant to this study, the authors should state so.

b.     However, there have been other papers that are related to Bayesian analysis for building energy consumption data (albeit also old) not mentioned by the author, such as:

                                               i.     MacKay DJ. Bayesian nonlinear modeling for the prediction competition. Build Eng 1994;100(2):1053–62.

                                             ii.     Chonan Y. A Bayesian nonlinear regression with multiple hyperparameters on the ASHRAE II time-series data. ASHRAE Trans 1996;102:405–11.

                                            iii.     Shonder JA, Im P. Bayesian analysis of savings from retrofit projects. Build Eng 2012;118:367.

The authors are encouraged to consult recently published literature reviews.

c.     Alongside ASHRAE’s Guideline 14, another important guiding document for calibrated simulation is the International Performance Measurement and Verification Protocol (IPMVP). This should also be cited.

6.     P2 L41-43: Please explain why likelihood-based methods are less likely to overfit.

7.     P3 L131: “MCMC” is not defined at first use.

8.     Figure 3: In “Necessary building properties”, the items are not all self-explanatory. This is because some items are straightforward values (e.g. GFA) but some seem more complex like “system” or “material”. The authors should point out at this figure that Table 2 provides examples.

9.     Figure 3: What the symbols refer to in “Input variables” and “Process variables” are not all explained. Also, the other tables (Table 1 & Table 3) that provide the nomenclature should be interlinked with this figure in captions.

10.  P6 L161: Here and elsewhere, I suggest that the province be referred to as “Guangdong, China”, so that the location is clearer to a global audience.

11.  Table 2: The energy consumption data is not presented clearly.

a.     Why is the length of data worded as “2 x 8760”? Is the sample composed of two separate collection periods? If so, why?

b.     What is the form of the measured energy (i.e. electricity, chilled/hot water, natural gas, or anything else)?

c.     Please consider visualising the data or part of the data with figures (time series or scatterplots) so that the readers can have a sense of the data’s characteristics and quality.

12.  P6 around eq (1): What do the subscripts q1 and q2 stand for? In “10 x 2 x 8760”, what is the “10”?

13.  P6 before eq (1): In the sentence “there are 2 x 8760 samples in z”, it should be “2 x 8760 data points” instead. A sample is a collection of data points.

14.  P7 L163 and elsewhere: I believe by “database” the authors mean “dataset”.

15.  P8 L181-182: What do “d” and the subscript “M” stand for?

16.  P8 L185: The table number is missing.

17.  Table 3: What is the definition of orientation?

18.  Table 3: Why are number of days and number of hours variables instead of constants? Do the authors mean “day of year” and “hour of day”?

19.  P8 L189-192: Please explain the symbols used in equations (5) – (8).

20.  Table 4: Guideline 14 should be cited also in the caption of this table (even though it is already cited in the text).

21.  P10 L239: I would suggest separating the results and discussion to two separate sections as a more traditional structure.

22.  Figure 5: Which variable does each number represent?

23.  Table 5: Is there a reason that U-values, ventilation rate, and internal gain are input variables (they are taken as accurate, if I understand correctly) instead of calibrated variables? From personal experience, these variables are usually highly uncertain. Especially the ventilation rate, which also strongly affects the computed energy consumption.

24.  Figure 8: Consider using the same vertical scale for both before- and after-calibration subplots to highlight the improvement. I understand this would make it difficult to see the fluctuations across the months in the post-calibration plot, but because the numbers are very small, the details become trivial.

25.  Figure 9. Consider using the same vertical scale across all subplots to highlight the different performance for each month.

26.  P17 L323-324: The statement “a slight inaccuracy in several days” is vague. This should be quantified, otherwise it is not necessary.

Author Response

Dear Reviewer,

Please kindly see the attachment.

Best regards,

Reviewer 2 Report

The paper deals with Bayesian Calibration for Office Building Energy Prediction Model. The overall paper is well designed, structured and written. However, there are few comments to the authors to improve the quality of the research: 

1. What is the originality of the research? how this research is different from other which are already published ?

2. Review section seems very OLD ! most of references are not recent why? Authors should refresh the list of references with more recent publications 

Author Response

(The authors gave the same response as above.)

Author Response

(The authors gave the same response as above.)

Round 2

Reviewer 1 Report

The authors have revised and improved the manuscript with great care and efforts. The topic of this study is interesting and important to the industry. However, I cannot find this manuscript ready for publication without another round of major revision. Below are some high-level comments of my main concerns:

11. The English language of this manuscript is inconsistent and some sections are very poorly written. Because of this, I find it difficult to follow several parts of this manuscript (and in some cases the authors’ responses). Extensive language editing is required.

22.       The studied buildings are not presented clearly. Their parameters are only provided in normalised forms While this is an effective and standard way for analysis, it takes away the opportunity for the readers to have an intuitive understanding of the buildings.

33.       In the previous round of review I expressed my concern over the selection of calibration variables, especially the exclusion of the ventilation rate which is sometimes difficult to determine and highly influential to energy consumption. The authors claimed that the calibration parameters were selected based on the behaviour of the data instead of human experience, and are supposedly more objective. However, such contradictions against field experience may negatively affect the credibility of the proposed method. I speculate that the lack of influence of the ventilation rate may be caused by the regional idiosyncrasies of the building system design. This then loops back to the flaw that the building properties are not introduced clearly in this manuscript.

Author Response

Dear Reviewer,

Please kindly find the attachment.

Best regards,

Reviewer 3 Report

The manuscript is revised according to my comments. 

Author Response

Dear Reviewer,

Sincerely thank you for your careful reading and helpful comments.

Best regards,

Round 3

Reviewer 1 Report

The authors have revised the manuscript carefully to address my concerns. Below are my remaining comments:

1.     I find the authors’ response to my concern towards the exclusion of ventilation rate convincing. The authors should include some of that response including the citation to the discussion section.

2.     Abstract L12 and elsewhere: I find it unusual to shorten CV-RMSE to just CV. I would recommend using CV-RMSE or CV(RMSE) instead.

3.     Table 2: Units should be provided for GFA.

Author Response

(The authors gave the same response as above.)
